# Exploring chromosomal structural heterogeneity across multiple cell lines

Ryan R Cheng[1]\*, Vinicius G Contessoto[1,2], Erez Lieberman Aiden[1,3], Peter G Wolynes[1,4,5,6], Michele Di Pierro[1,7]\*, Jose N Onuchic[1,4,5,6]\*

[1]Center for Theoretical Biological Physics, Rice University, Houston, United States; [2]Brazilian Biorenewables National Laboratory - LNBR, Brazilian Center for Research in Energy and Materials - CNPEM, Campinas, Brazil; [3]Center for Genome Architecture, Baylor College of Medicine, Houston, United States; [4]Department of Chemistry, Rice University, Houston, United States; [5]Department of Physics & Astronomy, Rice University, Houston, United States; [6]Department of Biosciences, Rice University, Houston, United States; [7]Department of Physics, Northeastern University, Boston, United States

**\*For correspondence:**
ryan.r.cheng@gmail.com (RRC);
m.dipierro@northeastern.edu
(MDP);
jonuchic@rice.edu (JNO)

**Competing interests:** The authors declare that no competing interests exist.

**Abstract** Using computer simulations, we generate cell-specific 3D chromosomal structures and compare them to recently published chromatin structures obtained through microscopy. We demonstrate using machine learning and polymer physics simulations that epigenetic information can be used to predict the structural ensembles of multiple human cell lines. Theory predicts that chromosome structures are fluid and can only be described by an ensemble, which is consistent with the observation that chromosomes exhibit no unique fold. Nevertheless, our analysis of both structures from simulation and microscopy reveals that short segments of chromatin make two-state transitions between closed conformations and open dumbbell conformations. Finally, we study the conformational changes associated with the switching of genomic compartments observed in human cell lines. The formation of genomic compartments resembles hydrophobic collapse in protein folding, with the aggregation of denser and predominantly inactive chromatin driving the positioning of active chromatin toward the surface of individual chromosomal territories.

## Introduction

The 3D spatial organization of the chromosomes in the nucleus of eukaryotic cells appears to be cell-type-specific (*Rao et al., 2014*; *Dixon et al., 2012*; *Dixon et al., 2015*; *Rowley et al., 2017*; *Dekker and Heard, 2015*; *Yu and Ren, 2017*; *Tang et al., 2015*). What determines this cell- type-specific organization and how that organization relates to patterns of gene expression remain crucial questions in structural genomics.

DNA–DNA ligation experiments have revealed spatial compartmentalization, generally termed A/ B compartmentalization (*Lieberman-Aiden et al., 2009*), and CTCF-mediated loop domains. It was observed that the A compartment chromatin contains a larger amount of the expressed genes while the B compartment chromatin is less transcriptionally active. Similar A/B compartmentalization has been observed across human cell lines (*Rao et al., 2014*; *Dixon et al., 2012*; *Dixon et al., 2015*) as well as in other species (*Dixon et al., 2012*; *Rowley et al., 2017*; *Dudchenko and Shamim, 2018*; *Sexton et al., 2012*; *Eagen et al., 2015*; *Zhang et al., 2012*), suggesting that compartmentalization is a conserved feature of genome organization across evolution. While single-cell structures can be interrogated using proximity ligation assays (*Nagano et al., 2013*; *Stevens et al., 2017*; *Tan et al., 2018*), high resolution has so far only been achieved through ligation methods when the

experiments are performed over a large population of cells, thus averaging over the respective individual 3D structures.

Recent microscopy approaches have begun to reveal the 3D structures of segments of chromatin longer than a megabase at a spatial resolution on the nanometer scale (*Bintu et al., 2018*; *Boettiger et al., 2016*; *Nir et al., 2018*; *Beliveau et al., 2015*). These approaches not only allow for the quantification of pairwise and higher-order interactions between loci, but also allow some quantification of the structural variability in a population of cells. One consistent observation from the imaging approaches, as well as from single-cell DNA–DNA ligation experiments (*Nagano et al., 2013*; *Stevens et al., 2017*; *Tan et al., 2018*; *Finn et al., 2019*), has been the high degree of structural variability seen within an apparently homogeneous population of synchronized cells of a single-cell type. Despite this variability, well-defined cell-type-specific DNA–DNA ligation maps for the ensemble emerge after population averaging the single-cell results.

The high degree of structural variability observed for chromatin necessitates structural models that go beyond a single energetic basin; without the existence of a native structure, Elastic Network Models (*Atilgan et al., 2001*) are likely not appropriate. Polymer models (*Barbieri et al., 2012*; *Jost et al., 2014*; *Gürsoy et al., 2017*; *Brackley et al., 2016*; *Tjong et al., 2012*; *Nuebler et al., 2018*; *Zhang and Wolynes, 2015*; *Di Pierro et al., 2016*; *Wong et al., 2012*; *MacPherson et al., 2018*) that describe the process of chromosome organization have been proposed. In particular, the Minimal Chromatin Model (MiChroM) has been shown to accurately predict the population-averaged DNA–DNA ligation maps (*Di Pierro et al., 2016*; *Di Pierro et al., 2017*; *Di Pierro et al., 2018*; *Contessoto et al., 2019*). Chromosomes are described as polymers subject to interactions which depend on the chromatin biochemical composition and on the genomic distance separating any two loci (*Di Pierro et al., 2016*). Genomic distance-dependent interactions recapitulate the effect of motors acting along the DNA polymer and result in lengthwise compaction of chromatin. Interactions depending on chromatin biochemical composition recapitulate transient binding among chromosomal loci and result in the emergence of compartmentalization through a process of phase separation, in which chromatin of the same biochemical type preferentially co-localizes. The propensity toward phase separation for chromosomes of human lymphoblastoid cells can be reliably predicted using epigenetic marking data (*Di Pierro et al., 2017*), suggesting that the information contained within the 1D epigenetic marking patterns decorating the chromatin polymer is sufficient to predict the ensemble of 3D chromosome structures. A neural network called MEGABASE (*Di Pierro et al., 2017*) was trained to quantify the statistical relationship between the experimental sub-compartment annotations and the histone methylation and acetylation markings tracks, as assayed using chromatin immunoprecipitation data. Once trained, MEGABASE can be used to predict the compartmentalization patterns of a chromosome using a set of epigenetic ChIP-Seq tracks as the sole input. Combining MEGABASE and MiChroM, we are able to simulate the structural dynamics of chromosomes.

We first use the MEGABASE+MiChroM computational pipeline (*Di Pierro et al., 2017*) to predict the 3D ensemble of chromosomal structures for several well-studied cell types: HMEC, HUVEC, IMR90, K562, HeLa-S3, and H1-hESC. To test these simulated 3D ensembles, we then generate ensemble averaged simulated ligation maps that are compared directly to population-averaged DNA–DNA ligation maps (*Rao et al., 2014*; *Dixon et al., 2012*). For the cell lines IMR90 and K562, we also use energy landscape tools to analyze the structures obtained through diffraction-limited microscopy by *Bintu et al., 2018* for short ~2 Mb segments of chromatin and compare the experimental structural ensembles directly with the corresponding regions of the simulated chromosome 21 for IMR90 and K562. This comparison shows that not only the population averages but also the structural heterogeneity that is observed in human chromosomes in the interphase are consistent with our energy landscape model. Chromosomes do not adopt a single structure in the interphase, but rather, exhibit a high structural variability characteristic of a phase-separated liquid. We provide a detailed characterization of this structural heterogeneity for the experimentally imaged and simulated segments of chromatin using a collective variable commonly used to quantify structural similarity in protein folding theory. For a gene-rich chromatin segment, we uncover two dominant clusters of structures in both the experimental and simulated structural ensembles: closed structures and open dumbbell-like structures. The transition from a closed structure to an open dumbbell appears to be governed by a two-state process with an apparent free energy cost of about four times the effective information theoretic temperature. For a gene inactive segment, structural analysis reveals

highly disordered structures that lack domain boundaries. Additionally, we further examine the structural differences between whole chromosomes belonging to different cell types. The simulations show that inactive segments of chromatin move to the interior of the chromosome, while gene active chromatin moves to the chromosome surface. This effect appears to be driven by the favorable effective interactions between loci belonging to the B compartment, which forms a stable interior core; a phenomenon reminiscent of the hydrophobic collapse much studied in protein folding.

## Results and discussion

### A polymer model of chromatin based on epigenetic features captures chromosome organization across different cell types

We previously developed a computational pipeline that can predict the 3D ensemble of chromosome structures by using chromatin immunoprecipitation tracks for histone modifications as input (*Di Pierro et al., 2017*). This approach was successfully used to predict the 3D chromosome structures for human lymphoblastoid cells (GM12878) using the experimental ChIP-Seq tracks for 11 histone modifications (*Di Pierro et al., 2017*), that is H2AFZ, H3K27ac, H3K27me3, H3K36me3, H3K4me1, H3K4me2, H3K4me3, H3k79me2, H3K9ac, H3K9me3, and H4K20me1. Predicted chromosome structures for human lymphoblastoid cells (GM12878) were found to be consistent with both DNA–DNA ligation and fluorescence in situ hybridization (FISH) experiments (*Rao et al., 2014*). Here we generate predictions beyond GM12878 to other well- studied cell lines for which we have found sufficient epigenetic marking data.

Using the MEGABASE neural network, which was previously trained using data from GM12878, and sourcing from the Encyclopedia of DNA Elements (ENCODE) database the ChIP-Seq tracks for the same 11 histone modifications previously used, sub-compartment annotations for all the autosomes of cell lines were generated that had never been used in the training phase of the neural network. These sequences of sub-compartment annotations, or chromatin types, then serve as input for molecular dynamics simulations using the Minimal Chromatin Model (MiChroM) (*Di Pierro et al., 2016*). Using this combined approach, the chromosomal structural ensembles for six additional cell lines were generated: human fetal lung cells (IMR-90), human umbilical vein endothelial cells (HUVEC), immortalized myelogenous leukemia cells (K562), human mammary epithelial cells (HMEC), human embryonic stem cells (H1-hESC), and HeLa-S3 cells.

For each cell type, averaging the simulated ensemble generates in silico DNA–DNA ligation maps, which are consistent with those determined experimentally. *Figure 1* shows the comparison between simulated and experimental maps for IMR90 (*Figure 1A*), HUVEC (*Figure 1B*) and K562 (*Figure 1C*), demonstrating quantitative agreement. Corresponding comparisons of the compartmentalization patterns are also provided in *Figure 1—figure supplements 1–3* for additional cell types HMEC, H1-hESC, and HeLa-S3, as well as for GM12878 in *Di Pierro et al., 2017*. In particular, the Pearson's $R$ between the simulated and experimental maps of matching cell type as a function of genomic distance shows that the long-range patterns of compartmentalization are captured over tens of mega-bases. To establish a term of comparison we calculated the Pearson's $R$ between the experimental DNA–DNA ligation maps of mismatching cell types. While the experimental observations on different cell lines do correlate with each other, computational modeling delineates the difference between cell type and appears to best match the experimental map when the cell types of simulation and experiment are matched up. This last result demonstrates that the present theoretical model discriminates well between different cell lines. Further, the Pearson's $R$ as a function of genomic distance demonstrates high quantitative agreement for matching cell types, comparable to the agreement between two biological replicates for the GM12878 ligation maps (*Rao et al., 2014*).

Additional comparisons between the experimental and simulated maps are shown in *Figure 1—figure supplement 5*. In particular, the scaling of the contact probability with genomic distance (*Figure 1—figure supplement 5*) appear to suggest that the chromosomes are denser in experiment than in the simulations for the cell lines HUVEC and H1-hESC. It is important to note that the simulations are not re-trained for the different cell line predictions; rather, all of the simulations are performed with a chromatin volume fraction of 0.1 (See Materials and methods for more details).

While we have focused so far on the spatial organization of entire chromosomes on the micrometer length scale, for a better comparison with the structures of chromosome 21 of IMR90 and K562

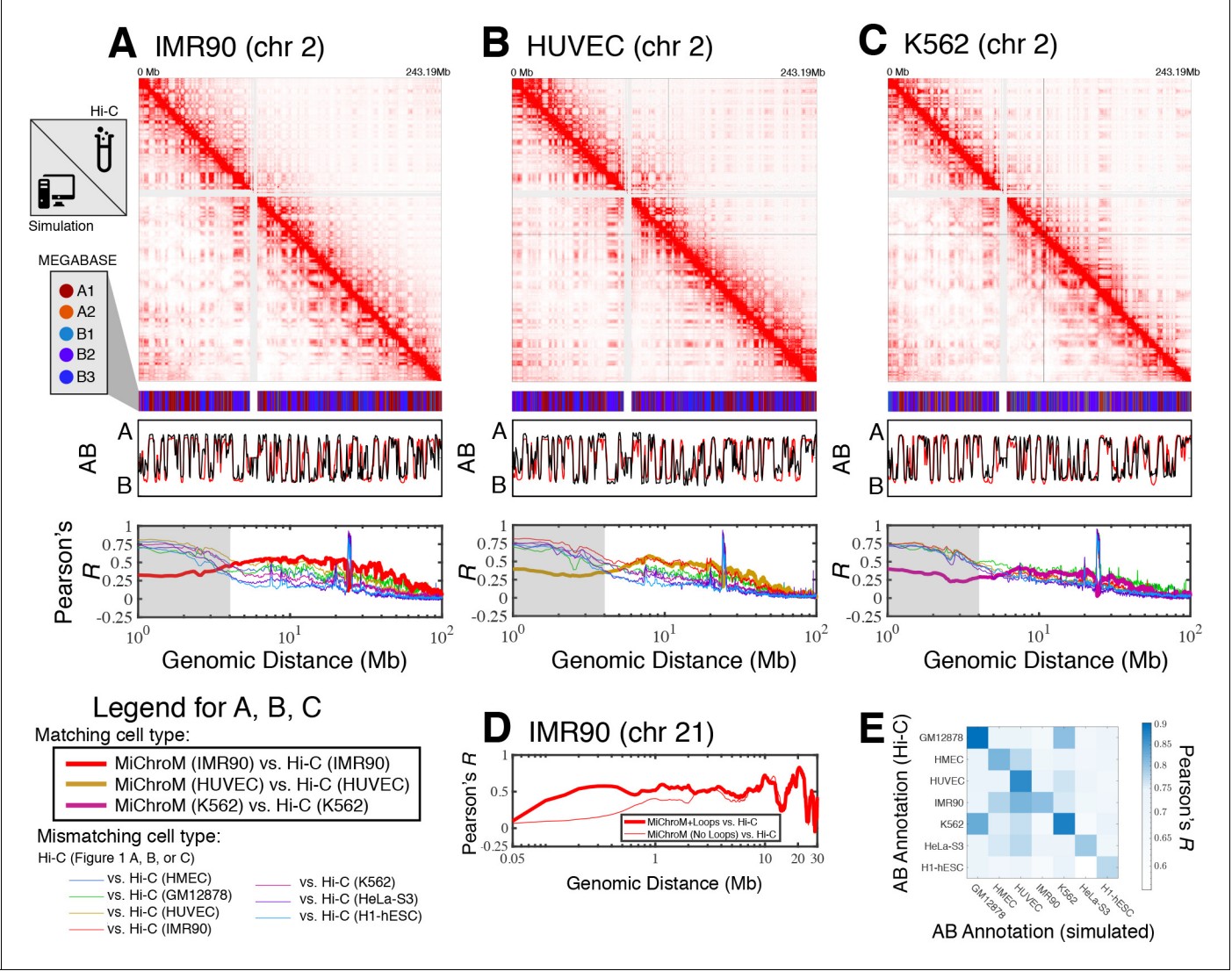

**Figure 1.** Prediction of chromosome structures for differentiated cell lines and for immortalized leukemia cells. The 3D ensemble of chromosome structures was predicted for the cell types (A) IMR90, (B) HUVEC, and (C) K562 using the ChIP-Seq histone modification tracks for the respective cell lines found on ENCODE—shown are the structural predictions for chromosome 2. As validation, the chromosome structures were compared with the DNA–DNA ligation experiments of *Rao et al., 2014*, where the simulated map is shown on the bottom left triangle and the experimental map is shown on the top right triangle. The datasets are visualized using Juicebox (*Durand et al., 2016*). The MEGABASE chromatin type annotation is shown as a color vector under the contact probability map, followed by the A/B compartment annotation (*Rao et al., 2014*) for the simulated map (red) and the experimental map (black), respectively. The Pearson's *R* between the simulated and experimental contact maps for fixed genomic distances are plotted for the cell types IMR90, HUVEC, and K562, respectively, in thick lines. The Pearson's *R* between the experimental maps of mismatching cell types are also shown with thin lines—See Legend. The shaded region highlights that at relatively short genomic distances (<10 Mb), excluding CTCF-mediated loops from the simulation results in disagreement between the simulated and experimental maps. When loops are included in the simulations, the agreement between the simulation and experiment is recovered at the short genomic distances. (D) Pearson's *R* as a function of genomic distance is plotted between the experimental map for chromosome 21 (IMR90) and MiChroM simulation with loops (thick red line) and without loops (thin red line). (E) A matrix of Pearson's *R* between the AB annotation of the experimental ligation map and the simulated contact maps for different cell types, respectively. The high Pearson's *R* signifies the consistency between the simulated maps and the experimental DNA–DNA ligation maps. Additional comparisons between simulated and experimental DNA–DNA ligation maps are shown for cell lines HMEC, H1-hESC, and HeLa-S3 in *Figure 1—figure supplements 1–3*, respectively. A matrix of Pearson's *R* between the AB annotation of the experimental ligation maps for different cell types is shown in *Figure 1—figure supplement 4*.

The online version of this article includes the following figure supplement(s) for figure 1:

**Figure supplement 1.** Prediction of chromosome structures for HMEC.

**Figure supplement 2.** Prediction of chromosome structures for H1-hESC.

**Figure supplement 3.** Prediction of chromosome structures for HeLa-S3.

*Figure 1 continued on next page*

*Figure 1 continued*

**Figure supplement 4.** A matrix of Pearson's *R* between the AB annotation of the experimental ligation maps for different cell types.

**Figure supplement 5.** Comparison of the experimental and simulated DNA–DNA ligation maps: power law scaling and scatter plot.

obtained from microscopy (*Bintu et al., 2018*), we have also incorporated in the polymer physics simulation the loops mediated by the activity of the protein CTCF.

*Figure 1D* shows that the inclusion of CTCF-loops, which are easily be incorporated into the model, improves the quality of the results for the short range features of chromosome organization within 10 Mb in genomic distance; at larger length scales the model appears to be completely insensitive of CTCF-mediated loops. To date, we have only examined the effect of the absence or presence of loops on the chromosome structural ensemble. A more detailed treatment of the short ranged chromatin contacts, particularly of many-body chromatin interactions (*Perez-Rathke et al., 2020*), has been shown to be important in recapitulating the organization of short segments of chromatin between 500 KB and 1.9MB in length.

*Figure 1E* shows the Pearson's *R* between the AB annotation vectors derived directly from the DNA–DNA ligation maps and those obtained from MiChroM simulations for different cell types. The diagonal of *Figure 1E* corresponds to the Pearson's *R* between AB annotations derived from experiment and simulation of matching cell types. The simulated and experimental annotations for the same cell types agree well with each other. *Figure 1—figure supplement 4* shows the Pearson's *R* between AB annotations derived from experiment alone for the different cell types. Notably, the high degree of correlation between the myelogenous leukemia cell line K562 and human lymphoblastoids (GM12878) maps observed in *Figure 1E* is apparent from DNA–DNA ligation maps alone (*Figure 1—figure supplement 4*). The agreement between the simulated and experimental A/B annotations is the highest quality (Pearson's *r* ~ 0.9) for the DNA–DNA ligation maps of GM12878, which is not surprising since the GM12878 has an order of magnitude more reads than any other map and consequently has the highest resolution.

Taken together, these results demonstrate that long-range compartmentalization observed in the DNA–DNA ligation maps is well captured by the simulated structural ensembles for these well-studied cell lines using only information about the epigenetic marking patterns as input.

## Chromatin structural ensembles from DNA-tracing reveal coexistence of open and closed structures

Recent developments in DNA-tracing have allowed the direct experimental determination of three-dimensional structures using diffraction-limited and super-resolution microscopy (*Bintu et al., 2018*; *Boettiger et al., 2016*; *Nir et al., 2018*; *Beliveau et al., 2015*). DNA-tracing is a technique that labels consecutive stretches of DNA with optical probes, which can be used to spatially resolve the positions of those probes using microscopy. It has become increasingly clear that unlike the situation for folded globular proteins, which typically can be reasonably well approximated for many purposes by a single native structure corresponding to the average conformation, chromatin appears to be highly dynamical and cannot be characterized by any single conformation. The heterogeneity of the chromosomal structural ensembles was first suggested by the analysis of the free energy landscape of chromosomes (*Zhang and Wolynes, 2015*; *Di Pierro et al., 2016*) and has been indirectly observed through single-cell DNA–DNA ligation experiments (*Nagano et al., 2013*; *Stevens et al., 2017*; *Tan et al., 2018*; *Finn et al., 2019*). The heterogeneity has now been confirmed by direct imaging of individual chromosomal structures (*Bintu et al., 2018*; *Boettiger et al., 2016*; *Nir et al., 2018*). As a consequence of this conformational plasticity, statistical ensembles (*Zhang and Wolynes, 2015*; *Di Pierro et al., 2016*; *Di Pierro et al., 2017*; *Di Pierro et al., 2018*; *Zhang and Wolynes, 2016*; *Di Pierro, 2019*; *Goundaroulis et al., 2020*; *Bascom et al., 2019*; *Dekker et al., 2013*; *Kalhor et al., 2012*) must be used in order to describe chromosomal structures in vivo.

In order to improve our understanding of the genomic structural ensembles, we characterize the structural heterogeneity of chromatin that was imaged using microscopy. We focus on the traced structures of *Bintu et al., 2018*, who obtained hundreds of images structures for short ~2 Mb segments of chromatin belonging to chromosome 21. These regions are 29.37–31.32 Mb (referred to here as Segment 1) of IMR90 and K562 cell types and 20.0–21.9 Mb (referred to as Segment 2) of

IMR90. Only structures where the positions of over 90% of the loci were resolved are used in the present analysis. There are then 692 usable structures for IMR90 Segment 1, 244 usable structures of K562 Segment 1, and 752 usable structures of IMR90 Segment 2.

As previously reported (*Bintu et al., 2018*; *Boettiger et al., 2016*; *Nir et al., 2018*), the traced structures can be used to generate a population-averaged contact maps, which turn out to be consistent with DNA–DNA ligation maps. Shown in *Figure 2—figure supplement 1* are the averaged contact maps for the chromatin Segments 1 (IMR90 and K562) and Segment 2 (IMR90), respectively. Nevertheless, information is lost when converting from a 3D structural ensemble to a 2D contact map.

Focusing on the structural details that cannot be found in a contact map, we make a close examination of the types of structures observed in the tracing dataset using a collective variable commonly used in studying protein folding landscapes, $Q$, which quantifies the structural similarity between two structures $\alpha$ and $\beta$ (*Eastwood and Wolynes, 2001*):

$$Q^{\alpha\beta} = \frac{1}{N} \sum_{i<j} exp\left( -\left( \frac{\left( r_{ij}^{(\alpha)} - r_{ij}^{(\beta)} \right)^2}{2\delta^2} \right) \right) \tag{1}$$

where $r_{ij}^{(\alpha)}$ and $r_{ij}^{(\beta)}$ are the distances between chromatin loci $i$ and $j$ in structures $\alpha$ and $\beta$:, respectively, $N$ is the number of pairs of loci included in the summation, and $\delta = 0.165 \mu m$ is the resolution length scale for which deviations in the distances between structures $\alpha$ and $\beta$ are treated as being similar. The $Q$ between any two structures ranges from 0 (dissimilar) to 1 (identical) over the entire set of pairwise distances between loci. The parameter $Q$ is not solely based on contacts; a pair of chromatin loci can contribute to $Q$ even if they are not spatially proximate if they are separated in both structures by a similar distance as set by $\delta$. In this way, $Q$ measures structure more stringently than a simple contact map does.

Using $1 - Q$ to define the distance between any two structures, hierarchical clustering of the traced structures for Segment 1 was applied to identify clusters having distinct structural features. These cluster sub-ensembles can be considered distinct conformational states. To see whether the Segment 1 structures for IMR90 and K562 exhibit a high degree of structural similarity, we combined their datasets before clustering.

When applied to the 936 combined experimental structures for Segment 1, the clustering algorithm yields three distinct clusters. These correspond to a closed dumbbell (Cluster 1), an open dumbbell (Cluster 2), and a highly dense chromatin state (Cluster 3) shown in *Figure 2*. The closed dumbbell, where the head and tail globular domains are in contact with one another, is the dominant state observed for Segment 1 in both IMR90 and K562, accounting for 97.4% of the imaged structures ($N_{closed} = 912$). Cluster 1 can further be sub-divided into subgroups 1a, 1b, 1 c, and 1d (*Figure 2*), which account for 75.5% of the structures in Cluster 1. The subgroups appear to capture various stages of the process of opening. The structures in subgroup 1b are fully collapsed, while structures in 1a, 1c, and 1d capture the progressive opening of the closed dumbbell. The distribution of the radius of gyration for structures belonging to sub-clusters 1a-1d is shown in *Figure 2— figure supplement 2*. The open dumbbell structures where the head and tail domains have dissociated from one another, account for approximately 1.8% of the imaged data ($N_{open} = 17$). Additionally, seven dense, highly compact structures were identified from clustering. Representative structures from the three clustered structural groups are shown in *Figure 2* and the corresponding population-averaged contact maps are shown in *Figure 2B and C* for the closed and open structures, respectively.

The high density chromatin, cluster 3, which was found when imaging both Segment 1 and Segment 2 (*Bintu et al., 2018*), is characterized by an extraordinarily high density of DNA $\sim 2 \times 10^3 mg/ml$, as estimated for naked dsDNA. For comparison, the density of heterochromatin that is estimated using microscopy data is $\sim 200 mg/ml$ (*Imai et al., 2017*); for this reason, we believe that these chromatin conformations are likely artifacts of the experimental protocol. We therefore have excluded Cluster three from further analysis.

Assuming that the opening of the chromatin Segment 1 is in an effective thermodynamic equilibrium would imply a relative stability of $log\left( N_{closed}/N_{open} \right) = E_{open} - E_{closed} \sim 4k_BT$, where $E_{open} - E_{closed}$ is

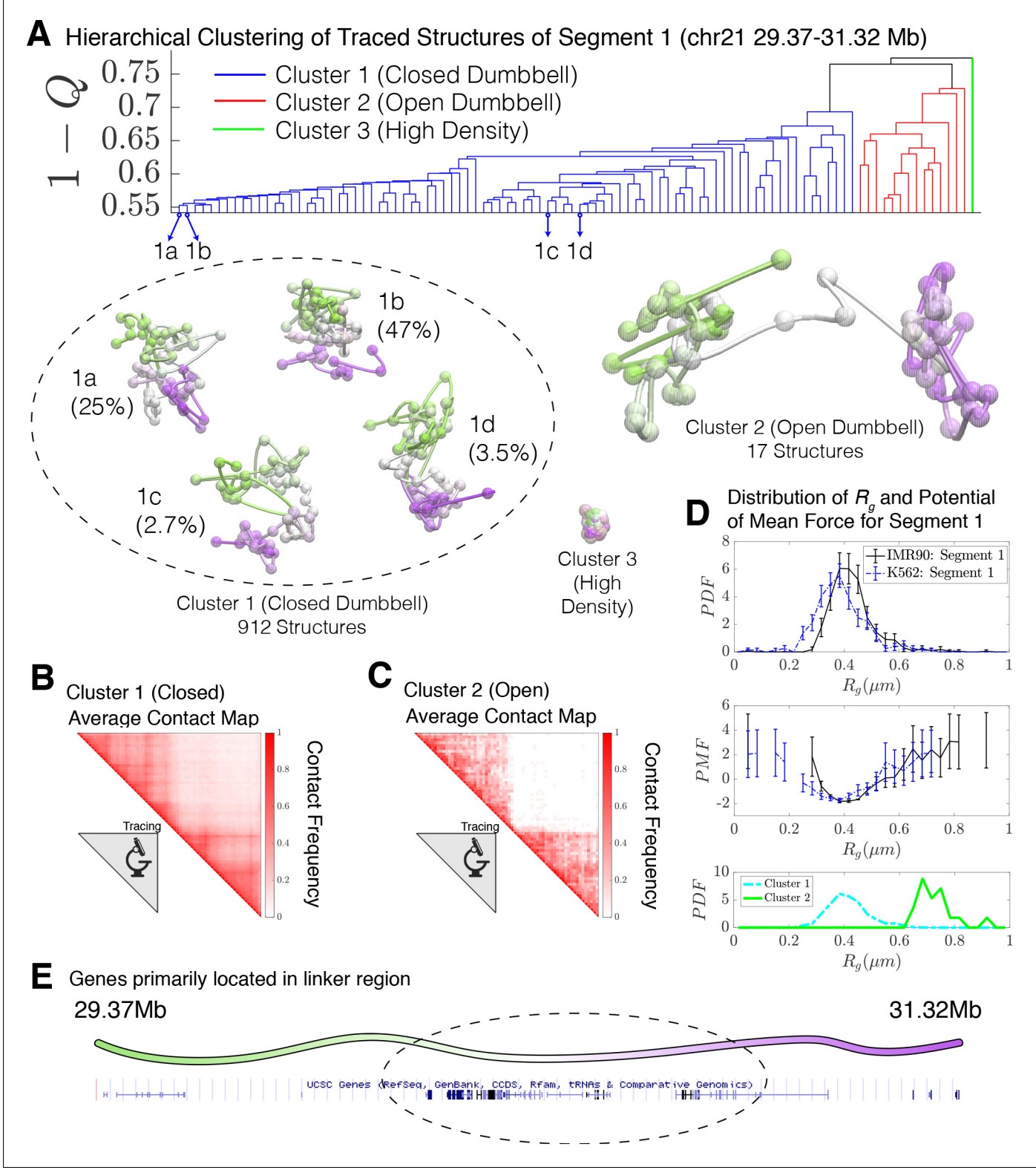

**Figure 2.** Hierarchical clustering and the detailed structural analysis of traced Segment 1. (**A**) The dendrogram representation of the hierarchical clustering of Segment 1 (chr21 29.37–31.32 Mb for IMR90 and K562 of *Bintu et al., 2018*), where $1 - Q$ is used as the distance between two structures. The clustering reveals three main clusters: closed dumbbell, open dumbbell, and highly dense structures. Further analysis of Cluster 1 reveals the presence of sub-clusters labeled 1a–1d that represent the gradual opening of the closed dumbbell. Representative traced structures are shown for each of the clusters and sub-clusters. The population-averaged contact maps for the closed and open structure clusters are shown respectively in (**B**) and (**C**),
*Figure 2 continued on next page*

*Figure 2 continued*

where 330 nm is used to define a contact between two 30 kb loci. (**D**) The distribution of the radius of gyration (top), the corresponding potential of mean force (center), and the distributions of radius of gyration for Cluster 1 and Cluster 2 (bottom) are shown for the traced structures of Segment 1 of IMR90 and K562. The distribution exhibits a heavy tail to the right of the average value, indicating the existence of open, elongated structures. (**E**) The UCSC Genes track is plotted along the genomic positions of Segment 1 using the Genome Browser (*Kent et al., 2002*). *Figure 2—figure supplement 1* shows the contact maps for the experimentally traced segments of chromatin. *Figure 2—figure supplement 2* shows the distributions of the radius of gyration for the sub-clusters of closed dumbbell structures obtained experimentally using tracing. *Figure 2—figure supplement 3* shows the hierarchical clustering and detailed structural analysis of the experimentally traced Segment 2.

The online version of this article includes the following figure supplement(s) for figure 2:

**Figure supplement 1.** Contact maps for the experimentally traced segments of chromatin.
**Figure supplement 2.** Distribution of radius of gyration for sub-clusters of closed dumbbell structures obtained experimentally using tracing.
**Figure supplement 3.** Hierarchical Clustering and the detailed structural analysis of traced Segment 2.
**Figure supplement 4.** The positioning of genes along traced Segment 1 and Segment 2.

an apparent free energy difference between the closed and open states and $T$ is an information theoretic temperature characterizing the ensemble (*Zhang and Wolynes, 2017*). Interestingly, the relative number of open and closed structures found in the simulations (discussed in the next section) is in remarkable agreement with this experimental finding.

We then used the radius of gyration , $R_g$, as an additional order parameter for the structural ensembles of Segment 1 belonging to IMR90 and K562 (*Figure 2D*). A corresponding potential of mean force can be extracted from the distribution of $R_g$ as $PMF = -k_B T log P(R_g)$, which also shows the free energy difference of $\sim 4k_B T$ between the closed (Cluster 1) and open (Cluster 2) structural sub-ensembles. The distributions of $R_g$ are also shown for Clusters 1 and 2 in *Figure 2D*. The open conformations (Cluster 2) possibly belong to a free energy minima in the PMF located between between $R_g \sim 0.6 - 0.8 \mu m$, although additional statistics would be necessary to firmly establish the presence of this additional conformational mode. Interestingly, the vast majority of genes appear to be positioned along the linker region connecting the two globular domains (*Figure 2E*).

Unlike Segment 1, Segment 2 of IMR90 completely lacks loop domains and, consequently, the averaged contact maps for Segment 2 exhibit no additional features beyond the decay in contact probability as a function of genomic distance (*Figure 2—figure supplement 1*). Structural analysis reveals that, without the presence of loop domains, Segment 2 is highly disordered; while clustering reveals open and closed structures, the lack of loop domains and domain boundaries results in the loss of dumbbell-like structures (*Figure 2—figure supplement 3*). It should be noted that unlike Segment 1, Segment 2 has an absence of genes (*Figure 2—figure supplement 4*).

## The chromosomal structures obtained from physical modeling are consistent with those observed with microscopy

We compare the chromosome structures sampled in the simulations to the diffraction-limited microscopy structures of *Bintu et al., 2018*, finding that the conformational states observed using microscopy are also found in the simulated structural ensemble without any calibration or fine tuning of parameters. While MEGABASE+MiChroM, provides us with structures of entire chromosomes, we focus specifically on the same ~2 Mb chromatin segment within chromosome 21 for our direct comparison.

It is important to note that the simulated model, and the structural variability that it captures, was derived from the energy landscape learned from population-averaged DNA–DNA ligation data using the principle of maximum entropy (*Di Pierro et al., 2016*). MiChroM has been shown to be consistent with experimental ligation maps (*Figure 1* and *Di Pierro et al., 2016*; *Di Pierro et al., 2017*; *Contessoto et al., 2019*), as well as the distribution of distances between Fluorescence in situ hybridization (FISH) probes (*Di Pierro et al., 2017*) and several observations regarding chromatin dynamics (*Di Pierro et al., 2018*).

Using the $1 - Q$ as the distance between all simulated structures for Segment 1, we now performed hierarchical clustering of the simulated structures. The dendrogram of this clustering is shown in *Figure 3A*, which uncovers two main clusters in the structural ensemble: a closed dumbbell (Cluster 1) and an open dumbbell (Cluster 2). The closed and open structures are consistent with those observed in the *Bintu et al., 2018* datasets. The representative structures of the closed and

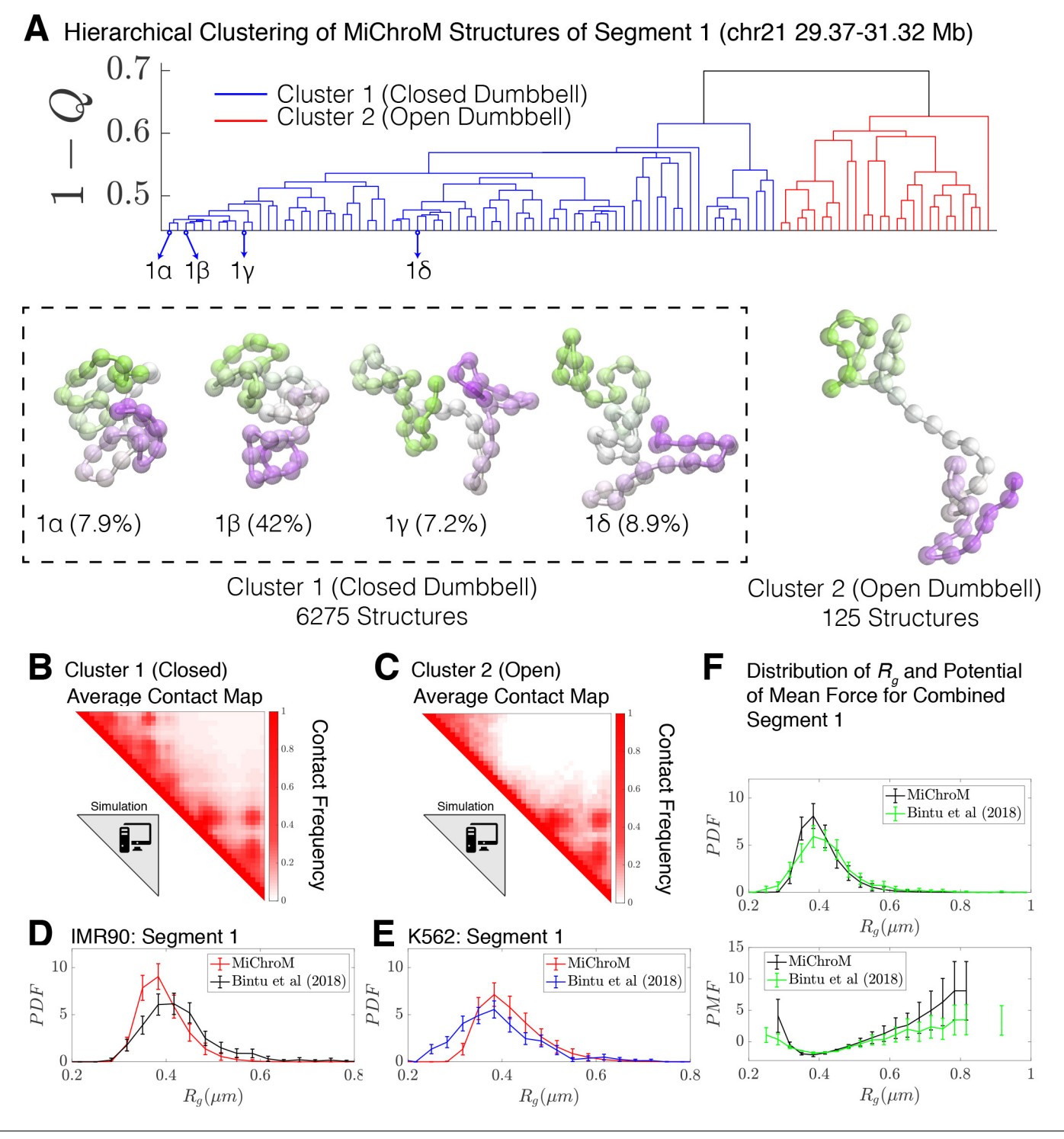

**Figure 3.** Hierarchical Clustering and the detailed structural analysis of simulated chromatin segment. (**A**) The dendrogram representation of the hierarchical clustering of simulated Segment 1 (chr21 29.37–31.32 Mb for IMR90 and K562) where $1 - Q$ (**Equation 1**) is used as the distance between two structures. The clustering reveals two main clusters: closed dumbbell (6275 out of 6400 structures) and open dumbbell (125 out of 6400 structures). The closed dumbbell can be subdivided into sub-clusters labeled $1\alpha$–$1\delta$ that represent the opening transition of the closed dumbbell. Representative structures are shown for each of the clusters and sub-clusters. The population-averaged contact maps for the clusters are shown respectively in (**B**) and (**C**), where 330 nm is used to define a contact between two 50 kb loci of the MiChroM model. The distribution of the radius of gyration is shown for Segment 1 IMR90 (**D**) and K562 (**E**) traced structures in comparison with the experimental structures. (**F**) Distribution of the radius of gyration and the corresponding potential of mean force is shown for both experiment and simulation for all of the structures of Segment 1. *Figure 3—figure*

*Figure 3 continued on next page*

*Figure 3 continued*

*supplement 1* shows the distributions of radius of gyration for the sub-clusters of closed dumbbell structures obtained from simulation. *Figure 3—figure supplement 2* shows how minor deviations in the unit of length estimate can account for the differences in the experimental and simulated distributions of radius of gyration .

The online version of this article includes the following figure supplement(s) for figure 3:

**Figure supplement 1.** Distribution of radius of gyration for sub-clusters of closed dumbbell structures obtained from simulation.

**Figure supplement 2.** Deviations in the unit of length estimate can account for the differences in the experimental and simulated distributions of radius of gyration .

**Figure supplement 3.** Comparison of the population-averaged contact maps from experimental tracing and simulation for Segment 1.

open conformations are shown in *Figure 3*, alongside the averaged contact maps for each of the clusters (*Figure 3B–C*), which are consistent with those determined experimentally (Shown in *Figure 2B–C*; *Figure 3—figure supplement 3*). The simulated Cluster 1 can again further be sub-divided into subgroups; 1$\alpha$, 1$\beta$, 1$\gamma$, and 1$\delta$ represent the four most populated sub-groups (*Figure 3*), which comprise 66% of the simulated structures of Cluster 1. The subgroups appear to capture various stages of the process of opening. The structures in subgroup 1$\alpha$ are fully collapsed, while structures in 1$\beta$, 1$\gamma$, and 1$\delta$ capture the progressive opening of the closed dumbbell. The radius of gyration of sub-clusters 1$\alpha$−1$\delta$ are shown in *Figure 3—figure supplement 1*.

No highly dense structures exist in the simulations. Such structures would collapse the entire chromatin segment to the volume of a single monomer, an occurrence that is prohibited by the energy function used to model the system. This is in harmony with our view that Cluster 3 seen in the experiments are artifacts of some sort.

For Segment 1, we performed our analysis on a set of 6400 structures, a representative subset of the simulated trajectories by taking every 125,000$^{th}$ structure from simulations. Both closed ($N_{closed} = 6275$) and open structures ($N_{open} = 125$) were identified by the clustering algorithm. Since MiChroM assumes an effective equilibrium thermodynamics representation of chromosome structures and dynamics, we can quickly calculate the relative stability between closed and open structures in the simulated ensemble as $log\left(N_{closed}/N_{open}\right) = E_{open} - E_{closed} \sim 4k_BT$, where $E_{open} - E_{closed}$ is the effective free energy difference between the closed and open states. This free energy difference is remarkably consistent with the value estimated using only the experimentally traced structures in the preceding section.

Finally, we calculated the distribution of the radius of gyration ,$R_g$, for the experimetal traced structures of *Bintu et al., 2018* and for the simulated MiChroM structures for Segment 1 belonging to IMR90 and K562 (shown in *Figure 3D* and *Figure 3E* respectively). Using a length scale calibrated previously (*Di Pierro et al., 2017*) from a single FISH experiment of 0.165 μm yields excellent quantitative agreement between the experimentally observed structures and those predicted de novo from simulation. It is particularly remarkable that any discrepancies between the experimental and simulated datasets can in fact be captured within 5% error of our original length estimate (*Figure 3—figure supplement 2*). Similarly, *Figure 3F* shows the direct comparison between the distribution of $R_g$ for Segment 1 as well as the corresponding potential of mean force. We see then that MiChroM appears to reproduce the apparent free energy difference between open and closed structures found using the experimentally traced structures.

## Comparative analysis of the chromosomal structural ensembles of different cell lines: connecting the epigenetic markings of loci with their radial positioning within territories

The frequency of chromatin type annotations predicted by MEGABASE over different cell types is shown in *Figure 4A* as a stacked bar chart that represents the distribution of chromatin type annotations predicted for each locus of chromosome 2 over all of the cell types. It is evident that certain loci have similar epigenetic markings patterns in all the cell types that we examined, either by being generally transcriptionally active loci, thus likely belonging to the A compartment, or by being transcriptionally inactive B compartment loci. On the other hand, several segments of chromatin switch compartments between different cell types.

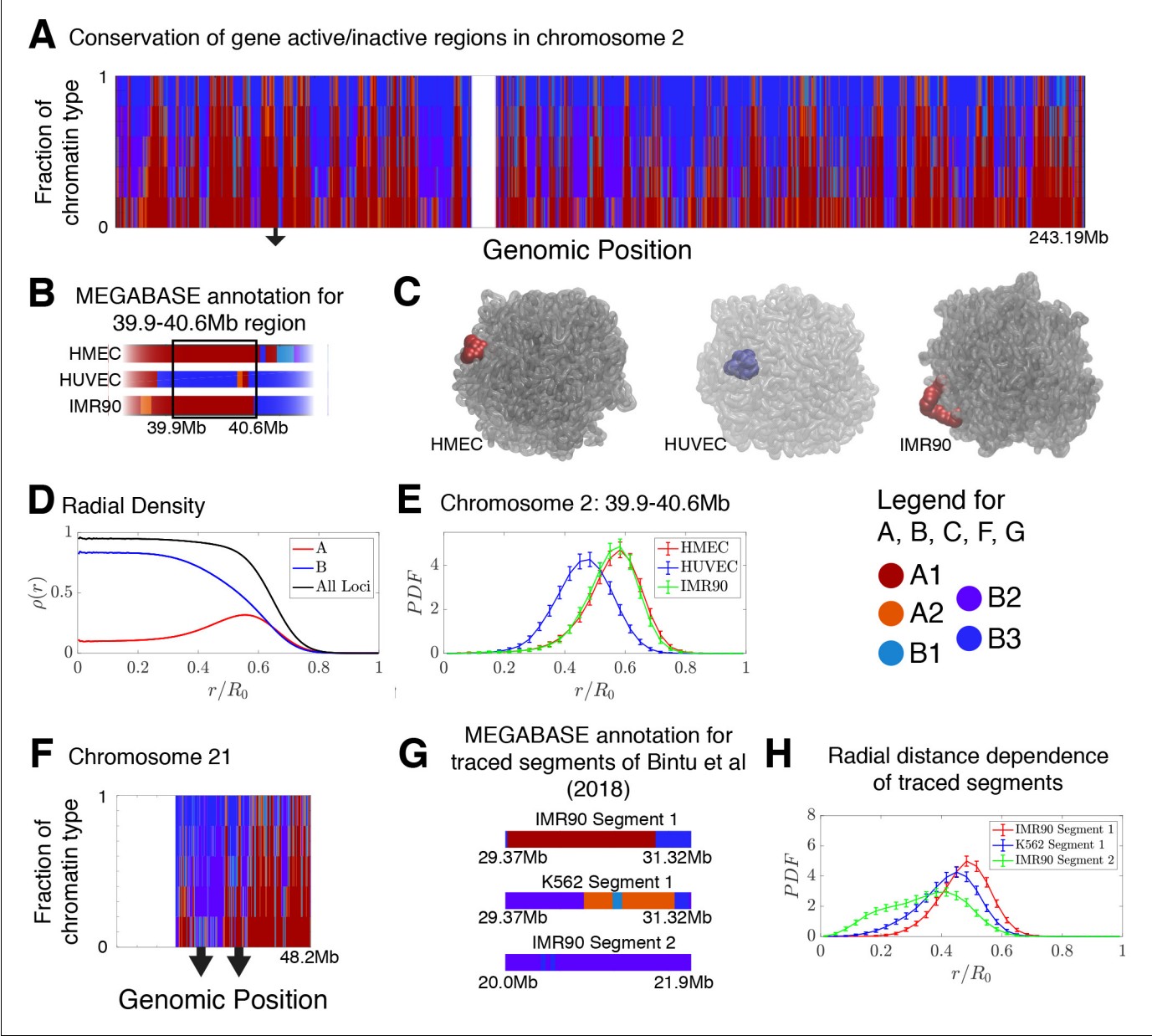

**Figure 4.** Conservation of compartmentalization across cell types and the radial dependence of marked chromatin. (**A**) A stacked bar chart is used to represent the distribution of chromatin type annotations predicted by MEGABASE as a function of the genomic position along chromosome 2 (hg19). The colors correspond the chromatin types given in the Figure Legend. For a given genomic position, the relative height of a particular color indicates the fraction of that particular chromatin type predicted at that locus. (**B**) The MEGABASE prediction of the chromatin type is shown for the chromatin segment 39.9–40.6 Mb of chromosome 2 for HMEC, HUVEC, and IMR90. A black arrow in (**A**) highlights the location of this segment. (**C**) The chromatin segment 39.9–40.6 Mb of chromosome 2 is shown in a representative structure for each of the cell types, where the color of the segment denotes its MEGABASE annotation. For HMEC and IMR90, the segment of chromatin tends toward the chromosome surface, whereas the segment tends toward the interior for HUVEC. (**D**) The radial density as a function of the normalized radial distance is plotted for A compartment loci, B compartment loci, and all loci for simulations of chromosomes for the HMEC cell type. (**E**) The probability density functions of the radial distance are shown for the center of mass of the segment 39.9–40.6 Mb of chromosome 2 for HMEC, HUVEC, and IMR90, respectively. (**F**) A stacked bar chart is used to represent the distribution of chromatin type annotations predicted by MEGABASE as a function of the genomic position along chromosome 21 (hg19). The arrows indicate the locations of the traced segments of *Bintu et al., 2018*: Segment 1 (29.37–31.32 Mb) and Segment 2 (20.0–21.9 Mb). (**G**) The MEGABASE annotation of the traced chromatin segments are given for IMR90 and K562. (**H**) The distribution of radial distances of the center of mass of each traced segment is shown.

Using the structural ensemble from the simulations based on the predicted compartments we then quantified the conformational differences between different cell types. On the chromosomal scale, structural differences emerge primarily from changes in the phase separation of epigenetically marked segments of chromatin. An example is illustrated in *Figure 4B*, which focuses on the region 39.9–40.6 Mb of chromosome 2 for HMEC, HUVEC, and IMR90. The MEGABASE classification (*Figure 4B*) identifies the segment in HMEC and IMR90 as belonging to the A compartment, whereas the segment for HUVEC should belong to the B compartment. Representative 3D structures for this segment for each of the respective cell types are shown in *Figure 4C*.

A plot of the radial density of A compartment loci and B compartment loci is shown in *Figure 4D*. These radial densities are consistent with previously reported simulations (*Di Pierro et al., 2016*). Taking a look at the radial distance of the center of mass of the segment of chromatin in each of the cell types, one finds that the A compartment loci tend to localize toward the surface of the chromosome, while the B compartment loci of the HUVEC cell type tend to localize in the interior (*Figure 4E*). A similar positioning of transcriptionally active chromatin toward the periphery of chromosomal territories was also observed by *Nagano et al., 2013* in mouse cells using Hi-C experiments.

We additionally use simulations to predict and examine the spatial positioning of the segments of chromatin examined by Bintu et al in the context of the entire chromosome 21. The experimental traced structures could not be used to ascertain the spatial positioning of those chromatin segments within chromosome 21 since only short segments were imaged rather than the entire chromosome. *Figure 4F* shows a stacked bar chart that represents the distribution of chromatin types predicted by MEGABASE for each genomic position of chromosome 21. *Figure 4G* shows the MEGABASE predictions for the traced segments, showing that IMR90 Segment 1 (29.37–31.32 Mb) is composed of A-type chromatin while IMR90 Segment 2 (20.0–21.9 Mb) primarily is composed of B compartment chromatin types. K562 Segment 1 (29.37–31.32 Mb) appears to contain both A and B chromatin types. *Figure 4H* shows the radial distance distribution of the center of mass of these segments of chromatin, showing that IMR90 Segment 2 tends to be in the interior, IMR90 Segment 1 tends to lie near the chromosome surface, and K562 Segment 1 occupies an intermediate region.

The finding that there exists a radial ordering associated with the spatial compartmentalization is consistent with the fact that according to the MiChroM potential (*Di Pierro et al., 2016*), contact interactions between B loci exhibit the most favorable energetic stabilization of all chromatin interactions. On the other hand, A to B or A to A-type interactions are comparably strong to each other, but are significantly weaker than the B to B interaction (See *Supplementary file 1*). In other words, according to the MiChroM energetic parameters (which were originally learned from Hi-C maps), B loci drive the phase separation of the chromosomes. Much like a hydrophobic-polar model from protein folding, the B compartmentalization forms the stable core of the simulated chromosome and the weaker interactions between A compartment loci with A or B loci tends toward the surface, to minimize the free energy of the molecular assembly. Our theoretical model thus corroborates recent experiments (*Falk et al., 2019*; *Strom et al., 2017*) that suggests heterochromatin phase separation is a major driving force behind genome organization, further highlighting the important role of phase separation in biological organization (*Di Pierro et al., 2016*; *Hnisz et al., 2017*; *Brangwynne et al., 2015*; *Banani et al., 2016*; *Shin and Brangwynne, 2017*).

## Discussion

DNA-tracing combined with diffraction-limited or super-resolution microscopy is beginning to shed light on the high degree of variability that is characteristic of chromosomal structures in the interphase (*Bintu et al., 2018*; *Boettiger et al., 2016*; *Nir et al., 2018*; *Beliveau et al., 2015*). These studies add to the growing body of evidence that a unique chromosomal fold simply does not exist in the interphase. Chromosome structures in the nucleus appear to be highly dynamical, owing to the many non-equilibrium processes in the cell, such as the activity of motor proteins.

The advances in genome imaging and the molecular simulation of chromosomes allows the development of parameters able to quantify the structural similarities between different chromosome structures, and the degree of heterogeneity in the ensemble of structures. Our results demonstrate that the collective variable Q, commonly used in protein folding studies and structural biophysics, is suitable for characterizing the structural ensemble of a segment of chromatin. Despite the high

degree of conformational plasticity, it appears that for segments of chromatin as short as the ones imaged by *Bintu et al., 2018* (~2 Mb in length), there do exist distinct clusters of chromatin structures that can be distinguished using Q. The dominant structures found for chromatin Segment 1 (chr21 29.37–31.32 Mb) examined using data from microscopy as well as from simulation can be described as being a closed dumbbell and an open dumbbell, where the ends of the dumbbell are the globular domains at the head and tail of the chromatin segment.

It is known that CTCF proteins bound along the genome act as gene insulators, probably through their suppressing activity toward loop extrusion (*Sanborn et al., 2015*; *Fudenberg et al., 2016*; *Vian et al., 2018*). Interestingly, a survey of the positioning of genes along Segment 1 reveals that the vast majority of genes appear clustered in the linker region of the chromatin segment (*Figure 2E*), sandwiched between the head and tail loop domains. On the other hand, there is an absence of genes located on Segment 2 (chr21 20.0–21.9 Mb) (*Figure 2—figure supplement 4*), which contains no loop domains. Classification of the experimentally imaged structures of Segment 2 lack the domain boundaries that segregate the head and the tail of the chromatin segment into globular domains, although still exhibiting open and closed conformations. These findings suggest a possible role in transcriptional regulation for the opening and closing of organized dumbbell structures. How open and closed structures would achieve such regulation of the transcriptional activity remains to be investigated. It is however clear that understanding the structure-function relationship in the genome is a crucial question that can only be answered using an accurate statistical characterization of the conformational ensembles.

Finally, our work refines the classical view of the spatial compartmentalization of chromatin. We find a striking dependence between radial positioning of chromatin and epigenetic marking patterns. Our theoretical model, MiChroM, predicts that transcriptionally active loci, typically belonging to the A compartment, move toward the surface of the chromosomal territory, while B compartment loci, typically inactive, move to the interior (*Di Pierro et al., 2016*). Since interactions among B-B loci result in the greatest energetic stabilization, aggregation of these loci seems to be driving force behind both the phase separation of epigenetically similar chromatin into compartments and the expulsion of the active chromatin toward the periphery of chromosomal territories. In other words, according to the present energy landscape model, when the epigenetic marking patterns of a locus are rewritten from A to B, the locus moves toward the interior of the chromosome, perhaps affecting the transcriptional activity of the associated genes.

## Materials and methods

We simulate the structural ensembles of chromosomes belonging to select human cell types using a previously introduced computational pipeline referred to as MEGABASE+MiChroM (*Di Pierro et al., 2017*). This pipeline takes chromatin immunoprecipitation tracks as input and computationally generates an ensemble of 3D structures of individual chromosomes at a resolution of 50 kb per monomeric unit. While this approach was initially trained and validated for chromosomes belonging to human lymphoblastoid cells (GM12878), we demonstrate that this approach can be readily applied to chromosomes belonging to any cell type given epigenetic histone modification data.

### Megabase

Maximum Entropy Genomic Annotations from Biomarkers Associated to Structural Ensembles (MEGABASE) (*Di Pierro et al., 2017*) was trained to quantify the correlations between chromosome structural annotations (i.e., compartment annotations A1, A2, B1, B2, B3) with chromatin immunoprecipitation (ChIP-Seq) signals. This allowed for the inference of the chromatin types (compartment labels) for each 50 kb locus of chromatin, given information about the histone modifications present at that locus.

### Discretization of ChIP-Seq data tracks

Chromatin Immunoprecipitatin (ChIP-seq) data was downloaded from ENCODE (*Dunham and Kundaje, 2012*) for the cell lines explored in this manuscript: IMR90, HUVEC, K562, HMEC, H1-hESC, and HeLa-S3. We focused on 11 histone modification tracks: H2AFZ, H3K27ac, H3K27me3, H3K36me3, H3K4me1, H3K4me2, H3K4me3, H3k79me2, H3K9ac, H3K9me3, and H4K20me1. These

11 tracks were previously shown to contain sufficient information to predict the chromosome structural ensembles for GM12878 (**Di Pierro et al., 2017**).

For each chromosome, the ChIP-Seq signal is re-casted into the data tracks at 50 kb resolution. This is performed by summing the ChIP-Seq signal contained within each 50 kb locus respective of each experiment.

The integrated ChIP-seq signal for each 50 kb locus is assigned a discrete state ranging from 1 (low signal) to 20 (high signal). This is performed by creating a histogram for each experiment of the integrated signal for all of the 50 kb loci in the chromosomes of each cell type. All loci belonging to the top 5% of the distribution with the highest signal are assigned the highest signal state, that is 20. The remaining 19 signal states are defined by partitioning the remainder of the distribution linearly with respect to the signal strength; loci are assigned to those states according to their integrated signal.

## Prediction of chromatin structural types from ChIP-Seq data using MEGABASE

The inferred probabilistic model (**Di Pierro et al., 2017**) can be marginalized to predict the chromatin type for a given locus *l* when given the experimental ChIP-Seq measurements at loci *l-2*, *l-1*, *l*, *l+1*, and *l+2*:

$$CST(l) = argmax P\big(CST | Exp_{1,...,L}(l-2, l-1, l, l+1, l+2)\big) \tag{2}$$

where $L = 11$ is the number of epigenetic histone modifications used in this study and Exp is a vector of discretized ChIP-Seq signals for loci *l-2*, *l-1*, *l*, *l+1*, and *l+2*. This allows for the prediction of the chromatin type (CST) for a given chromatin locus, given the ChIP-Seq signals for the 11 histone modifications at that locus. For additional details on the MEGABASE model, refer to **Di Pierro et al., 2017**. The predicted sequences of chromatin types can readily be obtained from our server (https://ndb.rice.edu/MEGABASE) (**Contessoto et al., 2019**) for different cell types and tissues with available ChIP-Seq histone modification tracks from ENCODE. The user can also supply ChIP-Seq tracks to generate sequences of chromatin type annotations for chromosomes of an unspecified cell type.

## Minimal chromatin model (MiChroM)

The sequence of inferred chromatin types for each chromosome serves as input for our coarse-grained simulations of individual chromosomes using the Minimal Chromatin Model (MiChroM) (**Di Pierro et al., 2016**). MiChroM is a coarse-grained representation of individual chromosomes with the following potential energy:

$$U_{MiChroM}\left(\vec{r}\right) = U_{HP}\left(\vec{r}\right) + U_{type-type}\left(\vec{r}\right) + U_{loops}\left(\vec{r}\right) + U_{ideal}\left(\vec{r}\right) \tag{3}$$

where

$$U_{type-type}\left(\vec{r}\right) = \sum_{\substack{k \geq l \\ k,l \in \text{Types}}} \alpha_{kl} \sum_{\substack{i \in \{\text{Loci of Type } k\} \\ j \in \{\text{Loci of Type } l\}}} f\left(r_{ij}\right)$$

$$U_{loops}\left(\vec{r}\right) = \chi \cdot \sum_{(i,j) \in \{\text{Loops Sites}\}} f\left(r_{ij}\right)$$

$$U_{ideal}\left(\vec{r}\right) = \sum_{d=3}^{500} \gamma(d) \sum_{i} f\left(r_{i, i+d}\right)$$

and the probability of crosslinking between chromatin loci *i* and *j* is modeled as

$$f\left(r_{ij}\right) = \frac{1}{2}\big(1 + tanh\left[\mu\left(r_c - r_{ij}\right)\right]\big). \tag{4}$$

The first term $U_{HP}$ is a homo-polymer potential that describes the connectivity (bonds and angles) between monomers—the monomers here represent a 50 kb span of DNA. The second term $U_{type-type}$

describes the sequence-dependent interactions between pairs of monomers; this term captures the phase separation of chromatin loci into spatial compartments. The parameters $\alpha_{kl}$ describe the energetic stabilization when two loci of chromatin type $k$ and $l$ are spatially proximal. The third term $U_{loops}$ describes the interaction between loop anchors that stabilize a CTCF-mediated loop. The final term $U_{ideal}$ referred to as the Ideal Chromosome (*Zhang and Wolynes, 2015*; *Di Pierro et al., 2016*; *Zhang and Wolynes, 2016*) describes the translationally invariant local ordering in chromatin; a pair of chromatin loci in close proximity are stabilized by an energy $\gamma(d)$ that depends on the genomic distance between the loci pair, $d = |i - j|$. Although the ideal chromosome accounts for the loop-length dependent entropic effects as well as motor-driven processes acting along the chromatin polymer, the term remains agnostic regarding the precise mechanisms responsible for local ordering.

The parameters $\mu = 3.22$ and $r_c = 1.78$ were adjusted for the contact maps of GM12878 B-lymphoblastoid cells in dataset GSE63525 (*Rao et al., 2014*). The parameters $\alpha$, $\chi$, and $\gamma$ were iteratively trained (*Di Pierro et al., 2016*) to be consistent with the DNA–DNA ligation map of chromosome 10 of human lymphoblastoid cells (GM12878)(*Rao et al., 2014*). The resulting polymer model is confined in a hard wall sphere to approximately preserve the volume fraction of chromatin in the interphase of 0.1 (*Rosa and Everaers, 2008*). Here, we model individual chromosomes confined within a hard sphere that represents a chromosome territory rather than the nuclear envelope. It has been shown that the physical tethering of chromatin to the nuclear envelope via the nuclear lamina may play an important role during differentiation and development (*Solovei et al., 2013*). Further, the role of the nuclear lamina in genome organization has been computationally modeled (*Laghmach et al., 2020*; *Lee et al., 2017*; *MacPherson et al., 2020*). While we do not currently use an explicit representation of the lamina, it should be noted that the MiChroM model was trained on DNA–DNA ligation data and its energetic terms would implicitly account for the effect of the lamina on the intra-chromosomal organization of chromatin loci.

MiChroM considers five chromatin types A1, A2, B1, B2, B3 plus a non-specific type NA, which is used to label the centromere. The $\alpha$ parameters, which govern the type-to-type interactions, are given in the *Supplementary file 1*. MiChroM makes no assumptions about the physical nature of the interactions that lead to compartmentalization. While the mechanistic details behind the chromatin type interactions are not fully understood, recent work (*MacPherson et al., 2018*) has shown that the binding of HP1 to chromatin can lead to compartmentalization via the oligomerization of HP1 to bridge the nucleosomes.

The parameter $\chi$ governing the loop interactions is equal to $-1.612990$.

The ideal chromosome potential is given by:

$$\gamma(d) = \frac{\gamma_1}{log(d)} + \frac{\gamma_2}{d} + \frac{\gamma_3}{d^2}$$

with parameters $\gamma_1 = -0.030$, $\gamma_2 = -0.351$, $\gamma_3 = -3.727$.

The reduced MiChroM energy function used in this manuscript omits CTCF-mediated loops unless stated otherwise:

$$U_{MiChroM}\left(\vec{r}\right) = U_{HP}\left(\vec{r}\right) + \sum_{\substack{k \geq l \\ k,l \, \in \, \text{Types}}} \alpha_{kl} \sum_{\substack{i \in \{\text{Loci of Type} k\} \\ j \in \{\text{Loci of Type} l\}}} f(r_{ij}) + \sum_{d=3}^{500} \gamma(d) \sum_i f(r_{i,\,i+d}) \tag{5}$$

For comparison with the DNA-tracing structures of *Bintu et al., 2018*, simulations of chromosome 21 for cell types IMR90 and K562 with CTCF-mediated loops were generated using the full energy function of MiChroM.

## Langevin simulations

Langevin simulations of individual chromosomes at a resolution of 50 kb per monomeric unit were performed using the GROMACS (*Abraham et al., 2015*) molecular dynamics package. Initial structures were generated from linear chain at a starting temperature of $3.0\varepsilon/k_B$ and linearly cooled to a temperature of $1.0\varepsilon/k_B$ over $5 \times 10^6$ steps with a time step of $0.002\tau$, where $\tau$ and $\varepsilon$ are the units of time and energy for our model, respectively. Following equilibration, simulations were run at a constant temperature of $1.0\varepsilon/k_B$ for $20 \times 10^6$ steps with a time step of $0.001\tau$. All simulations were run

using a dampening coefficient of 1.0τ. A total of 40 replicate simulations were run for each chromosome simulation. The resultant simulation trajectories are available for download at the Nucleome Data Bank (https://ndb.rice.edu/Data).

## Simulated DNA–DNA ligation maps

The simulated contact probability $p_{ij}$ between chromatin loci $i$ and $j$ is calculated by taking the expectation value of the probability of crosslinking (*Equation 4*) over the ensemble of chromosome structures obtained from simulation (*Di Pierro et al., 2016*):

$$p_{ij} = \langle f(r_{ij}) \rangle = \int d\vec{r} f(r_{ij}) exp\left(-\beta U\left(\vec{r}\right)\right) / \int d\vec{r} exp\left(-\beta U\left(\vec{r}\right)\right). \tag{6}$$

Here, $p_{ij}$ is a matrix element of the simulated DNA–DNA ligation map.

## Notes

Unless explicitly stated otherwise, all genomic positions are reported using the positions of the hg19 assembly. All of the simulated chromosome structures discussed in this manuscript were deposited in the Nucleome Data Bank (NDB) (*Contessoto et al., 2019*) found at https://ndb.rice.edu.

## Acknowledgements

The authors would like to thank Ting Wu for helpful discussions. This work was supported by the Center for Theoretical Biological Physics sponsored by the National Science Foundation NSF Grant PHY-2019745. JNO was also supported by the NSF-CHE-1614101 and by the Welch Foundation (Grant C-1792). JNO is a CPRIT Scholar in Cancer Research sponsored by the Cancer Prevention and Research Institute of Texas. V.G.C. is a Robert A Welch Postdoctoral Fellow and was also funded by FAPESP (São Paulo Research Foundation and Higher Education Personnel: Grant 2016/13998-8), and CAPES (Higher Education Personnel Improvement Coordination: Grant 2017/09662-7). Additional support to PGW was provided by the D R Bullard-Welch Chair at Rice University (Grant C-0016). E L A was also supported by the Welch Foundation (Q-1866), an NVIDIA Research Center Award, a McNair Medical Institute Scholar Award, an NIH 4D Nucleome Grant (U01HL130010), an NIH Encyclopedia of DNA Elements Mapping Center Award (UM1HG009375), an USDA award (559-6040-8-001) and a Binational Israeli Foundation Award (2019276).

## Additional information

### Funding

| Funder | Grant reference number | Author |
|---|---|---|
| National Science Foundation | PHY-2019745 | Ryan R Cheng<br>Vinicius G Contessoto<br>Erez Lieberman Aiden<br>Peter G Wolynes<br>Michele Di Pierro<br>Jose N Onuchic |
| National Science Foundation | CHE-1614101 | Jose N Onuchic |
| Welch Foundation | C-1792 | Jose N Onuchic |
| Cancer Prevention and Research Institute of Texas | | Jose N Onuchic |
| Welch Foundation | | Vinicius G Contessoto |
| Sao Paulo Research Foundation and Higher Education Personnel | 2016/13998-8 | Vinicius G Contessoto |
| Higher Education Personnel Improvement Coordination | 2017/09662-7 | Vinicius Contessoto |
| D. R. Bullard-Welch Chair at Rice University | Grant C-0016 | Peter G Wolynes |

| | | |
|---|---|---|
| Welch Foundation | Q-1866 | Erez Lieberman Aiden |
| NIH Office of the Director | U01HL130010 | Erez Lieberman Aiden |
| NIH Office of the Director | UM1HG009375 | Erez Lieberman Aiden |
| NVIDIA Research Center Award | | Erez Lieberman Aiden |
| McNair Medical Institute Scholar | | Erez Lieberman Aiden |
| United States-Israel Binational Science Foundation | 2019276 | Erez Lieberman Aiden |
| USDA | 559-6040-8-001 | Erez Lieberman Aiden |

The funders had no role in study design, data collection and interpretation, or the decision to submit the work for publication.

### Author contributions

Ryan R Cheng, Conceptualization, Resources, Data curation, Software, Formal analysis, Supervision, Validation, Investigation, Visualization, Methodology, Writing - original draft, Project administration, Writing - review and editing; Vinicius G Contessoto, Resources, Data curation, Software, Methodology; Erez Lieberman Aiden, Resources, Data curation, Software, Visualization; Peter G Wolynes, Formal analysis, Investigation, Writing - original draft, Writing - review and editing; Michele Di Pierro, Conceptualization, Formal analysis, Investigation, Methodology, Writing - original draft, Writing - review and editing; Jose N Onuchic, Funding acquisition, Investigation, Writing - original draft, Writing - review and editing

### Author ORCIDs

Ryan R Cheng https://orcid.org/0000-0001-6378-295X
Vinicius G Contessoto https://orcid.org/0000-0002-1891-9563
Jose N Onuchic https://orcid.org/0000-0002-9448-0388

### Decision letter and Author response

Decision letter https://doi.org/10.7554/eLife.60312.sa1
Author response https://doi.org/10.7554/eLife.60312.sa2

## Additional files

### Supplementary files

• Supplementary file 1. MiChroM parameters for type-to-type interactions. The energetic parameters are provided in units of $\varepsilon$. We consider five chromatin types (A1, A2, B1, B2, and B3) and a non-specific type (NA).

• Transparent reporting form

### Data availability

All of the simulated chromosome structures have been deposited in the Nucleome Data Bank (https://ndb.rice.edu/Data).

The following datasets were generated:

| Author(s) | Year | Dataset title | Dataset URL | Database and Identifier |
|---|---|---|---|---|
| Cheng RR, Contessoto VG, Aiden EL, Wolynes PG, Pierro MD, Onuchic JN | 2020 | Cheng_etal_K562Loops_2020 | https://ndb.rice.edu/static/structures/download/Cheng_etal_K562Loops_eLife_2020/NDB_3Dgenome_Cheng_etal_K562Loops_eLife_2020.dat | Nucleome Data Bank, K562_Loops |

| Cheng RR, Contessoto VG, Aiden EL, Wolynes PG, Pierro MD, Onuchic JN | 2020 | Cheng_etal_IMR90Loops_2020 | https://ndb.rice.edu/static/structures/download/Cheng_etal_IMR90Loops_eLife_2020/NDB_3Dgenome_Cheng_etal_IMR90Loops_eLife_2020.dat | Nucleome Data Bank, IMR90_Loops |
|---|---|---|---|---|
| Cheng RR, Contessoto VG, Aiden EL, Wolynes PG, Pierro MD, Onuchic JN | 2020 | Cheng_etal_H1-hESC_2020 | https://ndb.rice.edu/static/structures/download/Cheng_etal_H1-hESC_eLife_2020/NDB_3Dgenome_Cheng_etal_H1-hESC_eLife_2020.dat | Nucleome Data Bank, Cheng_etal_H1-hESC_2020 |
| Cheng RR, Contessoto VG, Aiden EL, Wolynes PG, Pierro MD, Onuchic JN | 2020 | Cheng_etal_HUVEC_2020 | https://ndb.rice.edu/static/structures/download/Cheng_etal_HUVEC_eLife_2020/NDB_3Dgenome_Cheng_etal_HUVEC_eLife_2020.dat | Nucleome Data Bank, Cheng_etal_HUVEC_2020 |
| Cheng RR, Contessoto VG, Aiden EL, Wolynes PG, Pierro MD, Onuchic JN | 2020 | Cheng_etal_HMEC_2020 | https://ndb.rice.edu/static/structures/download/Cheng_etal_HMEC_eLife_2020/NDB_3Dgenome_Cheng_etal_HMEC_eLife_2020.dat | Nucleome Data Bank, Cheng_etal_HMEC_2020 |
| Cheng RR, Contessoto VG, Aiden EL, Wolynes PG, Pierro MD, Onuchic JN | 2020 | Cheng_etal_Hela-S3_2020 | https://ndb.rice.edu/static/structures/download/Cheng_etal_Hela-S3_eLife_2020/NDB_3Dgenome_Cheng_etal_Hela-S3_eLife_2020.dat | Nucleome Data Bank, Cheng_etal_Hela-S3_2020 |
| Cheng RR, Contessoto VG, Aiden EL, Wolynes PG, Pierro MD, Onuchic JN | 2020 | Cheng_etal_IMR90_2020 | https://ndb.rice.edu/static/structures/download/Cheng_etal_IMR90_eLife_2020/NDB_3Dgenome_Cheng_etal_IMR90_eLife_2020.dat | Nucleome Data Bank, Cheng_etal_IMR90_2020 |
| Cheng RR, Contessoto VG, Aiden EL, Wolynes PG, Pierro MD, Onuchic JN | 2020 | Cheng_etal_K562_2020 | https://ndb.rice.edu/static/structures/download/Cheng_etal_K562_eLife_2020/NDB_3Dgenome_Cheng_etal_K562_eLife_2020.dat | Nucleome Data Bank, Cheng_etal_K562_2020 |

The following previously published datasets were used:

| Author(s) | Year | Dataset title | Dataset URL | Database and Identifier |
|---|---|---|---|---|
| Bintu B, Mateo LJ, Su J, Sinnott-Armstrong NA, Parker M, Kinrot S, Yamaya K, Boettiger AN, Zhuang X | 2018 | IMR90_chr21-18-20Mb.csv | https://github.com/BogdanBintu/ChromatinImaging/tree/master/Data/IMR90_chr21-18-20Mb.csv | GitHub, IMR90_chr21-18-20Mb.csv |
| Bintu B, Mateo LJ, Su J, Sinnott-Armstrong NA, Parker M, Kinrot S, Yamaya K, Boettiger AN, Zhuang X | 2018 | IMR90_chr21-28-30Mb.csv | https://github.com/BogdanBintu/ChromatinImaging/tree/master/Data/IMR90_chr21-28-30Mb.csv | GitHub, IMR90_chr21-28-30Mb.csv |
| Bintu B, Mateo LJ, Su J, Sinnott-Armstrong NA, Parker M, Kinrot S, Yamaya K, Boettiger AN, Zhuang X | 2018 | K562_chr21-28-30Mb.csv | https://github.com/BogdanBintu/ChromatinImaging/tree/master/Data/K562_chr21-28-30Mb.csv | GitHub, K562_chr21-28-30Mb.csv |

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
