## [Decision Letter]

**Acceptance summary:**

Using machine learning and polymer physics simulations, Cheng and colleagues generated structural models of chromosomes that are broadly consistent with single-cell imaging and capture the key characteristics of genomic compartmentation. With continued increase of single-cell imaging data and improvement in accuracy for training and calibration of such computational modeling, this holds great potential in generating accurate and full-scale chromosome structures and revealing the underlying mechanism of epigenetic switching at an molecular level.

**Decision letter after peer review:**

Thank you for submitting your article "Exploring Chromosomal Structural Heterogeneity Across Multiple Cell Lines" for consideration by *eLife*. Your article has been reviewed by three peer reviewers, including Yibing Shan as the Reviewing Editor and Reviewer #1, and the evaluation has been overseen by Detlef Weigel as the Senior Editor. The following individuals involved in review of your submission have agreed to reveal their identity: Huafeng Xu (Reviewer #2); and Jie Liang (Reviewer #3).

The reviewers have discussed the reviews with one another and the Reviewing Editor has drafted this decision to help you prepare a revised submission.

Summary:

Cheng et al. combined two computational methods previously published by the authors – MiChroM and MEGABASE – to simulate structural models of chromatins across six cell lines. They demonstrated good correlation between the average pairwise segment-segment contacts in the simulated models and the contacts inferred from ligation experiments and found similar statistical properties in the simulated models and in the structures observed by super-resolution microscopy. A number of important findings have been obtained from this study. A key finding is the high structural variability and ensemble nature of chromatins and as a result a chromatin generally cannot be described by a single stable structural fold. Furthermore, the study suggests that the structural linker region connecting compact domains contains most of genes for a chromosome segment.

1) The authors showed that different epigenetics of different cell types lead to different distributions of A/B compartments, which further lead to active A compartments moving toward to the surface of chromosomes, and inactive B compartments with stronger self-attraction move to the interior. However, it is well-known that LAD attachment can have strong effects on chromatin phase separations, as shown experimentally by Solovei et al., 2013, on lamin, and more recently in phase-field modeling work of Lee et al., 2017, and Laghmach, Pierro and Potoyan, 2020. In fact, in some studies this appear to be a dominant factor. The authors may wish to clarify whether lamina is included in the model, and whether the effects of LAD association would affect the conclusion of the results. Can the author provides a physical explanation of the A-B compartmentation beyond the analogy to hydrophobic collapse, and the possible physical elements that might regulate the open-close transition.

2) Figure 1E showed that predicted and measured A/B compartments have generally good correlation, with R varying for different cell types. It will be useful to plot the correlation R for A/B predictions/measurement, and R for simulated/measured Hi-C maps together. Are these R(A/B) and R(heatmaps) correlated? If not strongly correlated, why? Knowing this is useful as it may shed light on understanding to what extent the physical interactions encoded in the model (e.g. B-B hydrophobic-like interactions) explain the Hi-C maps, and how sensitive the current chromatin folding model is to the accuracy of A/B compartments assignment. Further, can the author provides a physical explanation of the A-B compartmentation beyond the analogy to hydrophobic collapse, and the possible physical elements that might regulate the open-close transition?

3) The authors showed that inclusion of CTCF-loops improves agreement of simulation and measured Hi-C heatmaps. Is this property uniquely for CTCF? A recent study showed that a small set of driver interactions exists for many loci, not necessarily involving CTCF, where 15-35 judiciously selected specific driver interactions can fold loci of enhancers of 500KB-1.9MB, with excellent R (>0.95) between measured and simulated Hi-C at 5 KB (PMC6966897). The authors may wish to further elaborate on their finding, e.g. after conducting a control study where a randomly (or non-randomly) chosen non-CTCF loop is include/not included, and whether such effects are also observed.

4) Simulated Hi-C heatmaps are reported for 7 cell lines. What are the cell nuclei sizes of these cells, and whether this size difference is taken into account in the simulations, and in general, why this is or is not an issue. This is relevant, as different nuclear size may affect the overall folding landscape of chromatin, since the effects of nuclear size confinement is one of the most strong constraints of chromatin.

5.1) The comparison with single-cell imaging data of Binto et al. is very interesting. It will be helpful if the authors show more details with examples of the degree of agreement between simulation and imaging studies. For example, a side-by-side comparison of best examples of single-cell heatmaps of Euclidean distance/contact between Bintu et al. measured single cells and simulated chromatin single-cell conformations. This can be supplemented with an overall scatter-plot of correlations of the pair-conformations of simulated/imaged conformations. Recent studies have shown some success in reproducing single cell Dip-C data through modeling (PMID PMC6966897), but comparison with imaging data is currently lacking, and the authors' results may set the standard for the field.

5.2) The 3D coordinates are available for both the simulated and the experimental structural ensembles. Thus, in addition to analyses of macroscopic and statistical properties such as the distribution of *R_g_* and the probability of open/close conformations, direct structural comparison should be feasible, which can show that indeed some of the structures generated by the simulations are highly consistent with the experimental structures. Can we compute the average accessible surface area for each 50kb segment and compare between simulations and experiments? Can we plot the contact probability for each pair of segments in the simulations against that in the experimental structures?

5.3) In Figure 1, the simulated structures appear to have significantly more contacts than inferred from ligation experiments. Does this imply that the simulated structures might be too compact? Is this because the experiment sampling is more limited than the simulations? Or because there is some potential systematic discrepancy despite the high-level consistency between the simulation and the experiment. This should be discussed.

6) While the clustering of both imaging and simulations clearly showed the heterogeneity of chromatin, how are clusters defined? If clustered by a different order parameter, e.g. *Q* instead of *R_g_*, do we still have the same finding – I wonder if certain natural order emerges from clustering. If we cluster both the simulated and experimental structures together, do they mingle in the same clusters?

7) There is an interesting discussion of the two-state equilibrium model and the PMF estimated energy gap of 4KT (also in the eighth paragraph of the subsection “Chromatin structural ensembles from DNA tracing reveal coexistence of open and closed structures”). However, can imaging studies of Segment 1 of Bintu et al. really be thought as from thermodynamic equilibrium? Its experimental set-up may be quite complicated and it is unclear whether imaged cells are samples properly drawn from the equilibrium distribution. In fact, the authors stated that chromatins are highly dynamic and follow a non-equilibrium process (Discussion, first paragraph).

8) "The genomic distance-dependent interactions recaptures the effect of motors acting along the DNA polymer." There is also well-known loop length dependent entropic effects as well, which may be part of the observed genomic distance dependency. It will be helpful if the authors clarify that only the motor-driven active process are at play in their model, or a mixture of both factors, and perhaps other hidden unspecified events are effectively accounted for in their model? Also, a citation on the motor process here will be helpful.

9) The reported finding that the linker region connecting two compact domains contains most of genes for segment 1 is very nice. Is this observation general?

10) The finding reported provides guidance for further biophysical studies of chromatin, for example, simple approximations such as Gaussian network models for chromatin domains are unlikely to be successful in capturing the heterogeneity of chromatin. The authors may want to briefly discuss the Gaussian network model in light of their results.

Revisions expected in follow-up work:

There was agreement that the value of the work could be greatly enhanced if the authors used their model to address a question of biological interest. For example, can the authors use their structural models-which have much richer details than available from experiments-to shed light on the distinction between the structural differences across different cell types and the variations among individual cells within the same cell type? What do the structural models teach us about transcription regulation?

---

## [Author Response]

Revisions for this paper:1) The authors showed that different epigenetics of different cell types lead to different distributions of A/B compartments, which further lead to active A compartments moving toward to the surface of chromosomes, and inactive B compartments with stronger self-attraction move to the interior. However, it is well-known that LAD attachment can have strong effects on chromatin phase separations, as shown experimentally by Solovei et al., 2013, on lamin A/C, and more recently in phase-field modeling work of Lee et al., 2017 and Laghmach, Pierro and Potoyan, 2020. In fact, in some studies this appear to be a dominant factor. The authors may wish to clarify whether lamina is included in the model, and whether the effects of LAD association would affect the conclusion of the results. Can the author provides a physical explanation of the A-B compartmentation beyond the analogy to hydrophobic collapse, and the possible physical elements that might regulate the open-close transition.

We thank the reviewers for bringing up this excellent point. While modeling entire genomes is our future direction, we currently have only modeled individual chromosomes confined within a volume that represents a chromosomal territory. The confinement in our case is not the nuclear envelope and we do not currently have any explicit description of the nuclear lamina. It should be noted however that the MiChroM model was trained on DNA–DNA ligation data and its energetic terms would implicitly account for the effect of the lamina on the intra-chromosomal organization of chromatin loci. We agree that the nuclear lamina has been shown to play an important role during differentiation and development (Solovei et al). However, for interphase chromosomes in differentiated cell lines, we have demonstrated that a model without explicit lamina can generate ensembles of 3D chromosome structures that are consistent with DNA–DNA ligation and microscopy experiments. We have added text to the manuscript regarding the potential role of lamina, as well as the referenced citations.

The strength of this work is that we use a model that makes no assumptions about the physical nature of the interactions that lead to compartmentalization. The parameters of the model were simply trained to be consistent with the Hi-C maps from experiment.

Recent work [MacPherson, Beltran and Spakowitz, 2018] has demonstrated how molecular interactions such as the binding of HP1 to chromatin can plausibly lead to compartmentalization via the oligomerization of HP1 to bridge the nucleosomes. However, the specific details of the molecular interactions that give rise to phase separation in the genome are not fully understood. Also unclear is the interplay between compartmentalization and motor proteins acting along the polymer, such as extruding complexes or polymerases. This has been clarified in the Materials and methods section.

2) Figure 1E showed that predicted and measured A/B compartments have generally good correlation, with R varying for different cell types. It will be useful to plot the correlation R for A/B predictions/measurement, and R for simulated/measured Hi-C maps together. Are these R(A/B) and R(heatmaps) correlated? If not strongly correlated, why? Knowing this is useful as it may shed light on understanding to what extent the physical interactions encoded in the model (e.g. B-B hydrophobic-like interactions) explain the Hi-C maps, and how sensitive the current chromatin folding model is to the accuracy of A/B compartments assignment. Further, can the author provides a physical explanation of the A-B compartmentation beyond the analogy to hydrophobic collapse, and the possible physical elements that might regulate the open-close transition?

We were very much interested in different ways to assess the predictions of our simulations in comparison with the experimental Hi-C maps. We focused on the Pearson between the A/B annotations between the simulation and experiment maps as well as the Pearson between the simulation and experimental maps at fixed genomic distances. These two measures are more stringent for comparing the compartmentalization patterns observed in the Hi-C maps.

While calculation of *R* between the simulated and experimental maps was discussed in our previous works, this particular measure is sensitive to the depth of sequencing of the experiment. The experimental data are irregular in terms of depth of sequencing (and therefore sparsity), which confounds the comparison. Global features of the ligation map such as the AB annotations are less sensitive to experimental irregularities than local features such as the direct ligation counts. Further, the Pearson between the maps are all high quality (>0.9) but they do not appear to be correlated to Figure 1E for the aforementioned reasons.

While MiChroM makes no assumptions of the molecular details that give rise to the phase separation, recent work in the Spakowitz lab [MacPherson, Beltran and Spakowitz, 2018] has shown that HP1 binding to chromatin can lead to distinct phases that resemble euchromatin and heterochromatin.

3) The authors showed that inclusion of CTCF-loops improves agreement of simulation and measured Hi-C heatmaps. Is this property uniquely for CTCF? A recent study showed that a small set of driver interactions exists for many loci, not necessarily involving CTCF, where 15-35 judiciously selected specific driver interactions can fold loci of enhancers of 500KB-1.9MB, with excellent R (>0.95) between measured and simulated Hi-C at 5 KB (PMC6966897). The authors may wish to further elaborate on their finding, e.g. after conducting a control study where a randomly (or non-randomly) chosen non-CTCF loop is include/not included, and whether such effects are also observed.

This is an interesting suggestion however we have not yet done such an in-depth analysis of short ranged loops. Rather, we focused on the global architecture of entire chromosomes. To date, we have only examined simulations with or without the presence of these CTCF-mediated loops and seen its effects on the global architecture. Our findings simply show that the inclusion of CTCF-mediated loops is important for features on the Hi-C map that are less than 10Mb in genomic distance (Figure 1D). We have added the citation and modified the text accordingly to point out that a more detailed analysis, in particular of driver interactions, might be important for an improved understanding of the organization of short segments of chromatin.

4) Simulated Hi-C heatmaps are reported for 7 cell lines. What are the cell nuclei sizes of these cells, and whether this size difference is taken into account in the simulations, and in general, why this is or is not an issue. This is relevant, as different nuclear size may affect the overall folding landscape of chromatin, since the effects of nuclear size confinement is one of the most strong constraints of chromatin.

The reviewers highlight a very important issue. However, at present we are not in the condition of addressing this point due to the lack of suitable experimental data regarding the nuclear size and shape distributions. All of the simulations are performed at a constant DNA density that is consistent with what is found in literature [citation: Rosa and Everaers, 2008]. It is worth pointing out that, using this constant density, we did previously predict FISH-measured distances (Di Pierro et al., 2017); this al least indicates that the overall degree of compaction of chromatin is generally correct. We look forward to further refine our models as soon as there are additional experimental observations regarding the nuclear size and shape.

5.1) The comparison with single-cell imaging data of Binto et al. is very interesting. It will be helpful if the authors show more details with examples of the degree of agreement between simulation and imaging studies. For example, a side-by-side comparison of best examples of single-cell heatmaps of Euclidean distance/contact between Bintu et al. measured single cells and simulated chromatin single-cell conformations. This can be supplemented with an overall scatter-plot of correlations of the pair-conformations of simulated/imaged conformations. Recent studies have shown some success in reproducing single cell Dip-C data through modeling (PMID PMC6966897), but comparison with imaging data is currently lacking, and the authors' results may set the standard for the field.

Thank you for your encouraging comment. We did in fact try to push further the direct comparison between imaging and simulations. We explored the comparison of ensemble averaged quantities such as the Hi-C maps and explored *Q* and *R_g_*. However, the structures of chromatin are quite fluid and it is difficult (and perhaps impossible) to choose which structures are to be selected out of the ensemble to make a direct comparison. This is consistent with the observation and perspective that chromatin structures are like “snowflakes”, because no two are the same ([See: https://www.nature.com/articles/d41586-019-01426-w].) This is further supported by single-cell Hi-C and Dip-C experiments. The individual structures are quite different from one another and a direct comparison of structures with a scatter plot of the pair distances from simulation and experiment would simply indicate that the agreement is poor until we run our simulations longer to find a better match. Precisely because of this reason, the goal of this work was to go beyond a comparison of single structures and to provide a statistical characterization of the experimental and simulated data sets. The text has been amended to better convey this point.

5.2) The 3D coordinates are available for both the simulated and the experimental structural ensembles. Thus, in addition to analyses of macroscopic and statistical properties such as the distribution of R_g_ and the probability of open/close conformations, direct structural comparison should be feasible, which can show that indeed some of the structures generated by the simulations are highly consistent with the experimental structures. Can we compute the average accessible surface area for each 50kb segment and compare between simulations and experiments? Can we plot the contact probability for each pair of segments in the simulations against that in the experimental structures?

As mentioned in the previous response, no two structures are the same, confounding our direct analysis of structures. The surface area would be a very interesting quantity to compare and measure, which is directly related to buried or exposed regions of chromatin. However, for the experimental imaging data that is available, the surface area is not well defined. For example, the position of a locus in an image structure represents the mean position of the imaged locus. On the other hand, the contact probability for each pair of segments can easily be calculated; we thank the reviewers for the suggestion. We have included the scatter plot of the contact probability for each pair of segments between simulated and experimental structures in Figure 1—figure supplement 5.

5.3) In Figure 1, the simulated structures appear to have significantly more contacts than inferred from ligation experiments. Does this imply that the simulated structures might be too compact? Is this because the experiment sampling is more limited than the simulations? Or because there is some potential systematic discrepancy despite the high-level consistency between the simulation and the experiment. This should be discussed.

The reviewers bring up an excellent point. It is important to note that the DNA–DNA ligation maps of human lymphoblastoid cells (GM12878) [ Rao et al., 2014] are by far the most well-sampled Hi-C maps in existence with at least an order of magnitude more reads than the HiC maps of the other cell types discussed in this manuscript. While it might appear that the simulated maps are more compact from looking at the maps, this is not the case if we are to look at the average contact probability as a function of genomic distance, which we have added to Figure 1—figure supplement 5. For the majority of the cell types, the power law scaling of the contact probability is the same in the experiments and simulations. However, the average contact probability is notably higher in the experimental Hi-C maps for both HUVEC and H1-hESC with respect to the simulated maps; this can be seen in Figure 1—figure supplement 5 in both the power law scaling and the scatter plots of the contact probabilities. This could imply that the chromatin density is incorrect in these simulations. As mentioned previously, all of the simulations are performed at a constant DNA density that is consistent with Ref: [Rosa and Everaers, 2008]; we do not currently have any additional information regarding chromosomal density (as discussed in response to comment #4). Furthermore, the Minimal Chromatin model is not re-trained for each cell type.

The main text has been modified to better discuss these points.

6) While the clustering of both imaging and simulations clearly showed the heterogeneity of chromatin, how are clusters defined? If clustered by a different order parameter, e.g. Q instead of R_g_, do we still have the same finding – I wonder if certain natural order emerges from clustering. If we cluster both the simulated and experimental structures together, do they mingle in the same clusters?

This is a deep question which is complicated by the fact that both the ordering parameters and the clustering algorithm affect the observed clusters, and yet, our hope is to uncover the true, natural clusters present in the data. In our work, we used agglomerative hierarchical clustering using 1-Q parameter as a distance. Clusters are naturally defined in the hierarchical clustering scheme by the distances 1-Q that separate different structures and are represented in the tree diagram of Figures 2 and 3. We did not cluster by *R_g_*; although after clustering by 1 – *Q*, we did calculate *R_g_* of each sub-cluster. There are many different parameters that can potentially be used to characterize structural ensembles; regardless of whether or not *Q* is the best collective variable to characterize chromatin structures, we have demonstrated that it can capture important structural features in real and simulated chromatin structures as both open and closed structures are identified from the experimental and simulated datasets.

7) There is an interesting discussion of the two-state equilibrium model and the PMF estimated energy gap of 4KT (also in the eighth paragraph of the subsection “Chromatin structural ensembles from DNA tracing reveal coexistence of open and 228 closed structures”). However, can imaging studies of Segment 1 of Bintu et al. really be thought as from thermodynamic equilibrium? Its experimental set-up may be quite complicated and it is unclear whether imaged cells are samples properly drawn from the equilibrium distribution. In fact, the authors stated that chromatins are highly dynamic and follow a non-equilibrium process (Discussion, first paragraph).

Thank you for your encouraging comment. In these HiC experiments, cells are synchronized so to be (predominantly) in interphase, the state of cellular homeostasis in between two successive cell divisions. Interphase is certainly not a state of thermodynamic equilibrium but is a long-lived steady state. This is also evident from the fact that it is possible to measure a well-defined characteristic contact map for the interphase ensemble. Our approach aims to capture the quasi-equilibrium of interphase using the typical Boltzmann distribution. We have no a priori guarantee this is a reasonable approach; we do however clearly show that this approach is sufficient to explain a large amount of experimental observation. In fact, at present there is no indication that a non-equilibrium theory is necessary, or useful, in modeling interphase human chromatin at the resolution treated in our manuscript (50kb), or larger. At the finer scale of kilobases, or in more dynamic phases of the cell life, we would clearly need to address the problem of non-equilibrium. To further complicate the matter, as the reviewers point out, it is also unclear how much the experimental procedure (fixing the cell, hybridization, etc.) degrade the observed ensemble.

8) "The genomic distance-dependent interactions recaptures the effect of motors acting along the DNA polymer." There is also well-known loop length dependent entropic effects as well, which may be part of the observed genomic distance dependency. It will be helpful if the authors clarify that only the motor-driven active process are at play in their model, or a mixture of both factors, and perhaps other hidden unspecified events are effectively accounted for in their model? Also, a citation on the motor process here will be helpful.

Thank you for your comment. The ideal chromosome includes both standard entropic effects and motors. Indeed, the approach is agnostic as to the full set of mechanisms at play. We have clarified the issue in the text as well as added references to the ideal chromosome.

9) The reported finding that the linker region connecting two compact domains contains most of genes for segment 1 is very nice. Is this observation general?

Thank you for your encouraging comment. Unfortunately, there isn’t a lot of experimental imaging data yet; therefore, the observation that we made about one particular 2Mb chromatin locus cannot yet be generalized. The next few years will truly be exciting times as more and more experimental data is generated. We look forward for more tracing/imaging experiments that examine gene rich regions.

10) The finding reported provides guidance for further biophysical studies of chromatin, for example, simple approximations such as Gaussian network models for chromatin domains are unlikely to be successful in capturing the heterogeneity of chromatin. The authors may want to briefly discuss the Gaussian network model in light of their results.

Thank you for your comment. One of our aims was precisely to shine light on this issue. We agree that the Gaussian network models likely will not be able to capture the conformational plasticity that is observed in the microscopy data and we need to go beyond methods that describe a single basin in the energy landscape. We have amended the text to make this point.

Revisions expected in follow-up work:There was agreement that the value of the work could be greatly enhanced if the authors used their model to address a question of biological interest. For example, can the authors use their structural models-which have much richer details than available from experiments-to shed light on the distinction between the structural differences across different cell types and the variations among individual cells within the same cell type? What do the structural models teach us about transcription regulation?

This is a wonderful suggestion that aligns perfectly with our research agenda for the coming years. Uncovering the structure-to-function relationship of the genome is a key direction of our center and we look forward to comparing our models to the new experimental observations being made using microscopy as well as ligation methods. Currently, our models suggest that gene active segments of chromatin tend to organize at the surface of the chromosome territory, which is consistent with single-cell Hi-C experiments (Nagano et al., 2013).